# Human Sterols Are Overproduced, Stored and Excreted in Yeasts

**DOI:** 10.3390/ijms25020781

**Published:** 2024-01-08

**Authors:** Astrid Radkohl, Veronika Schusterbauer, Lukas Bernauer, Gerald N. Rechberger, Heimo Wolinski, Matthias Schittmayer, Ruth Birner-Gruenberger, Gerhard G. Thallinger, Erich Leitner, Melanie Baeck, Harald Pichler, Anita Emmerstorfer-Augustin

**Affiliations:** 1Institute of Molecular Biotechnology, Graz University of Technology, NAWI Graz, 8010 Graz, Austria; 2BioTechMed-Graz, 8010 Graz, Austria; 3Bisy GmbH, 8200 Hofstaetten an der Raab, Austria; 4Institute of Biomedical Informatics, Graz University of Technology, 8010 Graz, Austria; 5Department of Molecular Biosciences, University of Graz, NAWI Graz, 8010 Graz, Austria; 6Field of Excellence BioHealth, University of Graz, 8010 Graz, Austria; 7Institute of Chemical Technologies and Analytics, Technische Universität Wien, 1040 Vienna, Austriaruth.birner-gruenberger@tuwien.ac.at (R.B.-G.); 8Institute of Analytical Chemistry and Food Chemistry, University of Graz, NAWI Graz, 8010 Graz, Austria; erich.leitner@tugraz.at; 9Acib—Austrian Centre of Industrial Biotechnology, 8010 Graz, Austria

**Keywords:** cholesterol, ergosterol, *Komagataella phaffii*, *Pichia pastoris*, yeast, plasma membrane, lipid droplets, lipotoxicity, sphingolipids

## Abstract

Sterols exert a profound influence on numerous cellular processes, playing a crucial role in both health and disease. However, comprehending the effects of sterol dysfunction on cellular physiology is challenging. Consequently, numerous processes affected by impaired sterol biosynthesis still elude our complete understanding. In this study, we made use of yeast strains that produce cholesterol instead of ergosterol and investigated the cellular response mechanisms on the transcriptome as well as the lipid level. The exchange of ergosterol for cholesterol caused the downregulation of phosphatidylethanolamine and phosphatidylserine and upregulation of phosphatidylinositol and phosphatidylcholine biosynthesis. Additionally, a shift towards polyunsaturated fatty acids was observed. While the sphingolipid levels dropped, the total amounts of sterols and triacylglycerol increased, which resulted in 1.7-fold enlarged lipid droplets in cholesterol-producing yeast cells. In addition to internal storage, cholesterol and its precursors were excreted into the culture supernatant, most likely by the action of ABC transporters Snq2, Pdr12 and Pdr15. Overall, our results demonstrate that, similarly to mammalian cells, the production of non-native sterols and sterol precursors causes lipotoxicity in *K. phaffii*, mainly due to upregulated sterol biosynthesis, and they highlight the different survival and stress response mechanisms on multiple, integrative levels.

## 1. Background

With their flat, sturdy, steroid structures and the small hydroxyl head group, sterols are structurally different from other lipids found in biological membranes. Due to these unique structural features, sterols are considered as membrane reinforcers and dynamic regulators [1]. They modulate cell membrane fluidity, sustain membrane domain structures, and strengthen the barrier function between the cell and the environment [2]. Sterols are also essential for cell viability, which is why any imbalance in sterol homeostasis has an enormous impact on the growth and healthy development of all eukaryotic cells [3,4]. As a consequence, the exact cellular responses to changes in sterol compositions are difficult to study, with most experimental data obtained either from studies using sterol biosynthesis inhibitors [5,6] or from analyzing the few viable knockout strains [7].

Like most essential pathways, the sterol biosynthetic pathway is broadly conserved from lower to higher eukaryotes, with mammalian cells producing cholesterol and fungi producing ergosterol as the most dominant sterol species [8]. Yeast has been serving as a model organism to study disorders of sterol metabolism. For example, diverse *Saccharomyces cerevisiae* strains carrying deletions of non-essential ergosterol biosynthesis pathway genes were analyzed as sterol metabolic disease models (reviewed by [9]). Yeast was also exploited as a model for Smith-Lemli-Opitz syndrome (SLOS), which occurs when the terminal enzyme of cholesterol biosynthesis, 7-dehydrocholesterol reductase (DHCR7), is defective [10]. SLOS patients exhibit abnormal physical characteristics, possess below-average cognitive abilities and react negatively to sensory stimuli [11]. However, when knocking out late *ERG* genes (*ERG2–ERG6)*, the effect on cell physiology was not as severe as expected [12]. Possible explanations are that the structural differences between ergosterol and its immediate precursors are too small to disturb the cellular processes in yeasts to the same extent as in mammalian cells, or the simple fact that yeasts can adapt more easily to changes in sterol structures. It was also shown that cholesterol can be exchanged for ergosterol in mammalian cells, and vice versa in yeast cells, and still promote cellular growth [13,14]. A cholesterol-producing *Komagataella phaffii* (syn. *Pichia pastoris*) strain was also used to functionally produce mammalian Na^+^/K^+^-ATPase, which needs to physically interact with cholesterol for correct localization [15]. In contrast to late *ERG* mutations, cholesterol as the dominant sterol enormously affected cell physiology, including abnormal cell growth, the defective localization of diverse membrane proteins and hypersensitivity to different reagents. The same strain was shown to dramatically upregulate cell wall biosynthesis [16,17], an effect never reported for any *ERG* gene deletion strain.

In this study, we conducted an integrative multilevel analysis of cholesterol-producing *Komatagella phfaffii* and *S. cerevisiae* in order to globally investigate the cellular response mechanisms triggered by defective sterol biosynthesis. Multiple adaptions on the transcriptome, lipidome and proteome levels were observed, some of them posing important survival mechanisms of the lipotoxic effects caused by the sterol exchange, others resulting from the absence of correct sterol–protein or sterol–lipid interaction and, most likely, some stemming from secondary effects. One of our major findings was that cholesterol strains massively overproduce sterols, an extremely harmful condition typically called lipotoxicity [18]. In order to prevent lipotoxicity, cells usually buffer an excess of free sterols in the ER via esterification, followed by storage in lipid droplets (reviewed by [19]). Mammalian cells additionally have the ability to reduce their intracellular cholesterol load by excreting free cholesterol by ABC transporters ABCA1 and ABCG1 [20]. To date, no equivalent mechanism has been reported for yeasts. Instead, defective sterols were found to enter a lipid quality control cycle in which the sterol is reversibly acetylated by the alcohol acetyltransferase Aft2 and deacetylated by the membrane-anchored lipase Say1 [21]. Acetylated sterols are then excreted from cells, whereas Say1 determines which sterols are retained. The excretion of cholesteryl acetate requires ongoing vesicular transport mediated by the Pathogen-Related Yeast (PRY) proteins Pry1 and Pry2 [22], but is independent of the ABC transporters Aus1 and Pdr11, the main plasma-membrane-associated transporters responsible for sterol uptake [23]. Since yeasts are not able to catabolize sterols, the question remains as to how sterols, and especially free sterols, may ultimately leave the cells. In this study, we show that ABC transporters are responsible for efficient sterol excretion in *K. phaffii*, whereas acetylation is no prerequisite for these molecules to enter the excretory pathway. Altogether, our results demonstrate that cholesterol-producing yeast is an excellent reporter system to study sterol-related cellular processes, dysfunction and stress response mechanisms. 

## 2. Results

### 2.1. Exchange of Ergosterol for Cholesterol Impacts the Regulation of Diverse Lipid Biosynthesis Pathway Genes

Structurally, ergosterol differs only slightly from cholesterol in having two additional double bonds (at positions C7 and C22) and a methyl group at C24 of the side chain, and the production of cholesterol instead of ergosterol was achieved upon the deletion of *ERG5* and *ERG6* and overexpression of mammalian *DHCR7* and *DHCR24* [14] (Figure 1A). In order to compare the transcriptomics of a cholesterol-producing and wild-type *K. phaffii* strain, high-depth RNA sequencing data were generated. The raw reads were deposited in the EMBL-EBI database under accession number PRJEB70602. Of the 5291 protein-coding genes annotated, 5204 genes were sufficiently expressed in at least one condition. In total, we identified 1313 differentially expressed genes (DEGs; LFC (log fold change) > 1, false discovery rate (FDR) < 0.05). Of these, 719 genes were upregulated with LFCs up to 9.3, and 594 genes were significantly downregulated with an LFC in the expression level as low as −9.7 (Figure 1B).

Based on the results published by Adelanto et al. [24], a schematic representation of the lipid biosynthesis pathways from *K. phaffii* was created, including phospholipids, sphingolipids, sterols and fatty acids, and the fold changes of gene expression were assigned (Figure 2). While the genes of phospholipid biosynthesis were overall downregulated, ergosterol biosynthesis genes were strongly upregulated. *ERG2* and *ERG3*, as well as *ERG24* and *ERG25*, depicted the highest LFC values (Figure 1B and Figure 2). In sphingolipid biosynthesis, early pathway genes were up- and late genes downregulated in the cholesterol strain (Figure 2).

### 2.2. Integrated Transcriptomic–Lipidomic Effect of Cholesterol Production in K. phaffii

Whole cell extracts and purified plasma membrane fractions (quality control in Appendix A) of the wild-type and cholesterol strain were analyzed by LC-MS. The levels of phosphatidylcholine (PC), the most abundant phospholipid in the plasma membranes [25], remained similar in the cholesterol and wild-type strain (Figure 3). The levels of phosphatidylethanolamine (PE) and phosphatidylserine (PS), however, were found to be reduced by 50% in the cholesterol strain. Our transcriptome data of genes in the PS biosynthetic pathway revealed that the Cds1-activated conversion of phosphatidic acid (PA) to cytidine diphosphate diacylglycerol (CDP-DG) was downregulated (LFC = –1.18), whereas the expression of *CHO1*, which further converts CDP-DG to PS, was barely affected (LFC = –0.14) (Figure 2). PC, PS and PI are synthesized in microsomes, whereas the vast majority of PE is produced in mitochondria, which requires the efficient transport of its substrate, PS, to the inner mitochondria membrane for conversion by Psd1 [26]. While the expression of *PSD2* remained largely unchanged, *PSD1* was slightly downregulated with, an LFC of −0.62. Ups2 is another important protein involved in PE synthesis, which localizes to the mitochondrial intermembrane space and transfers phosphatidylserine from the outer to the inner mitochondrial membrane [27]. We found the transcription level of *UPS2* to be about threefold upregulated (LFC = 1.41). PE and PC are also synthesized by the Kennedy pathway [28], and the expression of Kennedy pathway genes (*CKI1*, *PCT1*, *ECT1*, *EPT1*) seemed largely unaffected by cholesterol production. However, the methylation of PE to PC was strongly downregulated. The levels of phosphatidylinositol (PI) were increased in the plasma membranes of the cholesterol strain, even though the biosynthesis of the predominant precursors, inositol and CDP-DG, was not dramatically upregulated. This may be explained by the fact that the further conversion of PI to PIP by the *STT4*-encoded PI kinase was downregulated (LFC = −1.28) in the cholesterol strain. The catabolism of phospholipids involves their hydrolysis into fatty acids and other lipophilic substances by phospholipases Plb1, Plb2 and Plb3 [29]. In *K. phaffii*, only the homolog for *PLB3* could be annotated so far, which is strongly downregulated (LFC = −2.02) in the cholesterol strain.

### 2.3. Fatty Acid Side Chain Saturation Is Changed in the Cholesterol Strain

We found that *K. phaffii* produces a diverse range of polyunsaturated fatty acids (Figure 3B). *S. cerevisiae*, in contrast, was reported to produce no polyunsaturated fatty acids (PUFAs) with more than two double bonds [30]. In the cholesterol *K. phaffii* strain, a shift towards polyunsaturated fatty acids was observed for most phospholipid species (PC, PE and PS) (Figure 3B), whereas side chain saturation varied from species to species. Transcriptome data revealed that the two delta(9) fatty acid desaturases *OLE1-1* (*FAD9A*) and *OLE1-2* (*FAD9B*) and the delta(12) fatty acid desaturase *ODE1* (*FAD12*) were upregulated, with LFCs of 1.73, 1.07 and 1.14, respectively. It is known that the fatty acid saturation of phospholipids plays an important role in regulating membrane fluidity, especially under stressful conditions like growth at elevated temperatures [31] or adaptation to hazardous reagents [32,33]. Hypothesizing that *K. phaffii* adapts fatty acid acyl chain saturation in order to counterbalance changes in membrane fluidity, we compared the growth of the wild-type and cholesterol strain at different temperatures. The cholesterol strain exhibited increased cold and heat sensitivity, whereas the growth defect was more severe at 37 °C (Figure 3C).

### 2.4. Sphingolipid Levels Are Affected by Production of Human Sterols

For the comparison of sphingolipid levels, whole cell extracts and PM fractions were analyzed. Cholesterol production dramatically influenced the sphingolipid levels and compositions in *K. phaffii* (Figure 4). In particular, the levels of hexyl ceramide (HexCer) (Figure 4A) and ceramide (Cer) (Figure 4B) were decreased, which can be explained by the downregulation of *LMT1* (−0.98) and, to a lesser extent, *GCS1* (−0.29), the two genes responsible for converting DHS-d4,8 to the respective end-products (Figure 2). On the contrary, the levels of IPC increased, while the levels of MIPC decreased (Figure 4C,D), probably due to the downregulation of *CHS1* (LFC −1.86), which converts IPC to MIPC. M(IP)_2_C could not be detected either in the wild type or the cholesterol strain with the extraction and detection method applied. Fatty acid elongases, involved in sphingolipid biosynthesis, act on very-long-chain fatty acids of up to 24 carbons in length. In *S. cerevisiae*, two *ELO* genes, *ELO1* [34] and its paralog *ELO2* [35], which arose from the whole genome duplication, exist. In *K. phaffii*, which did not undergo whole genome duplication, there was only one *ELO* gene, *ELO2*, which was downregulated (LFC −1.15) in the cholesterol strain.

### 2.5. Sterols and TAGs Are Produced in Excessive Amounts

Whole cell extracts of the *K. phaffii* (Figure 5A) and *S. cerevisiae* (Appendix A) wild-type and cholesterol strains were analyzed by thin layer chromatography (TLC). Diverse reference (standard) lipids relevant for this study were co-applied for rough identification. Band intensities were quantified using ImageJ. While the amount of free sterols was not significantly changed, sterol esters were enriched in the *K. phaffii* and *S. cerevisiae* cholesterol strains three- and twofold, respectively. TLCs also confirmed TAG accumulation in the *K. phaffii* cholesterol strain. By taking a closer look at the different TAG species, we found the enrichment of C52 and C54 (Figure 5B), which has been reported for *K. phaffii* before [34]. While the overall distribution of saturated and unsaturated TAG species was not drastically affected by the production of cholesterol, an overall increase in every single component was observed. TAGs are produced from diacylglycerol (DAG) and acetyl-CoA [35], and we also found all detectable DAG species to be enriched in the cholesterol strain (Figure 5C).

### 2.6. Cholesterol and TAGs Accumulate in the Cells

LDs are the major cellular organelles for the storage of neutral lipids. In order to assess the total LD mass, we stained cells with BODIPY 493/503. Fluorescent microscopy images showed a ~twofold increase in LD volumes in the cholesterol-producing *K. phaffii* and *S. cerevisiae* strains (Figure 6A), which was confirmed by the image-based quantification of fluorescent dots per total cell area (Figure 6B). We also observed a general difference between LDs found in *K. phaffii* and *S. cerevisiae*. While *K. phaffii* cells contained single, larger droplets, *S. cerevisiae* cells harbored many, but smaller droplets. For the comparison of cellular sterol levels, the *K. phaffii* and *S. cerevisiae* wild-type and cholesterol strains were analyzed by GC-MS (Figure 6C). For both yeasts, we noticed ~threefold increased amounts of sterols in the sterol-modified strain, and, according to TLC analysis, the increase could largely be attributed to sterol esters (Figure 5A). In the case of *S. cerevisiae*, cells mostly produced cholesterol, while *K. phaffii* produced a mixture of cholesterol and its precursors, 7-dehydrocholesterol (7-DHC), cholesta-8-24(25)-dienol and cholesta-5,7,24(25)-trienol (Figure 6C). The accumulation of 7-DHC and cholesta-8,24(25)-dienol indicates that *DHCR7* and *DHCR24* are not as active or efficiently expressed in *K. phaffii* on glucose as in *S. cerevisiae* [13]. Since we did not observe a drastic difference in the overall sterol amounts in the PM fractions of the wild-type and cholesterol-producing strain (Appendix A), excessive amounts of sterols seem to be largely stored in lipid droplets.

In order to find a mechanistic explanation for the accumulation of sterol esters and TAGs, we consulted the transcriptome data. The transcripts of *CDS1* (LFC–1.18), which is coding for the protein converting PA to CDP-DG for PS biosynthesis (see Figure 2 for the biosynthetic pathway), indicated the possible accumulation of PA. In addition, *DGK1*, a gene encoding for an ER-membrane localized diacylglycerol kinase that balances PA levels by converting DAG to PA, was downregulated (LFC −0.83). DAG is then further converted to TAG by Lro1 and Dga1. While *DGA1* was not differentially regulated, *LRO1* was downregulated (LFC −1.58). The most interesting finding was that both triacylglycerol lipases found in *K. phaffii*, *TGL3* (LFC −0.91) and *TGL4* (LFC −0.41), were downregulated, supposedly causing the accumulation of TAGs and lipid droplets in the cholesterol strain. A similar phenomenon was observed for sterol esters: while the conversion of free sterols to sterol esters by Are2 was not upregulated (LFC −0.26), the hydrolysis of sterol esters by Yeh2 was strongly downregulated (LFC −1.86).

### 2.7. Sterol Biosynthesis Is Transcriptionally Upregulated

In addition to LD metabolism (Figure 6D), we also investigated the early pathway of sterol biosynthesis (Figure 7A). First, we inspected acetyl-CoA formation, the central molecule for sterol and TAG production. Glycolysis generates pyruvate, which is then decarboxylated by pyruvate decarboxylase (PDC) to yield acetaldehyde. Coenzyme A biosynthesis requires the uptake of panthothenate from the culture supernatant by Seo1-1 (LFC 0.55) and Seo1-2 (LFC 1.00), which is then further converted to coenzyme A by the coenzyme A-synthesizing protein complex (CoA-SPC). The CoA-SPC consists of various Cab proteins, where the expression of diverse CAB genes was reported to be regulated by sterol regulatory elements in *S. cerevisiae* [37,38]. Both glycolysis and coenzyme A biosynthesis were transcriptionally upregulated in the cholesterol *K. phaffii* strain (Figure 7A). Additionally, *IME4*, which was only recently shown to stimulate glycolysis and acetyl-CoA synthesis in *S. cerevisiae* [39], was upregulated 2.3-fold in *K. phaffii* (LFC 1.16). Hence, there seems to be an overall increased flux through this early pathway in the cholesterol strain.

Next, in the acetyl-CoA biosynthetic pathway, acetaldehyde is subject to dehydrogenation, and CoA is ligated to generate acetyl-CoA under the catalysis of acetaldehyde dehydrogenase (ALD) and acetyl-CoA synthetase (ACS) [40]. In *K. phaffii*, we found two *ACS* genes, *ACS1* and *ACS2*. *ACS2* was found to be 2.5-fold downregulated, which may explain why we found ~60% less acetyl-CoA in the cholesterol strain (Figure 7B). The strong downregulation of *ACS2* and decreased levels of acetyl-CoA could result in the accumulation of acetate, and, indeed, acetate transporters *ADY2-2* (LFC = 0.84) and especially *ADY2-4* (LFC 7.31) were massively upregulated in the cholesterol strain. Even though we did not specifically investigate the acetate levels in the cells and cell supernatants, we found the supernatant of exponentially growing cholesterol cells to be more acidic (pH 4.5 ± 0.3) than that of wild-type cells (pH 5.6 ± 0.2).

After the conversion of acetate to acetyl-CoA, aceto-acetyl-CoA is formed by *ERG20* (LFC 0.31) and further converted to Hmg-CoA by *ERG13* (LFC 1.40) for sterol biosynthesis. The upregulation of *ERG13* (LFC = 1.40) led to a ~40% increase in Hmg-CoA in the cholesterol strain (Figure 7B). The expression of *ERG13* is positively regulated by the sterol regulatory element binding protein Upc2 in *S. cerevisiae* [41,42,43]. Upc2 is a transcription factor for most *ERG* genes and thus the main regulator of ergosterol biosynthesis in yeast. In *S. cerevisiae*, Upc2 has a paralog, Ecm22 [44], that arose from the whole genome duplication, but this gene has neither been annotated nor did we succeed in finding any Ecm22 homolog in *K. phaffii* BLASTp searches. Therefore, we focused on *UPC2*. *UPC2* was strongly upregulated (LFC 2) in the cholesterol strain, and we indeed also noticed the strong upregulation of *ERG2* and *ERG3* (Figure 2), two genes previously reported to be regulated by Upc2 [44]. Therefore, we deleted *upc2*∆ and investigated the intracellular sterol levels. Even though we did not succeed in deleting *upc2*∆ in the *K. phaffii* wild-type strain when applying different strategies, including CRISPR/Cas9, *UPC2* could easily be deleted in the cholesterol-producing strain. The deletion of *UPC2* decreased the sterol levels by ~25% (Figure 7C), which indicates that other mechanisms, besides transcriptional activation by *UPC2*, must upregulate sterol biosynthesis in the cholesterol strain.

### 2.8. Cholesterol Precursors Are Efficiently Exported by Pdr Transporters

It was reported that *S. cerevisiae* mainly excretes small amounts of acetylated sterols with the help of Pry2, when the gene encoding the deacetylase Say1 was deleted [22]. In the cholesterol strain, *PRY2* and its potential paralog, *EPX1* [45], were transcriptionally upregulated, with LFCs of 0.99 and 1.62, respectively. Immunoblot analysis of strains producing Pry2-HA and Epx1-3FLAG revealed higher titers of these proteins in the supernatant of the cholesterol strain (Appendix A). However, we were not able to identify any acetylated sterol species in the culture supernatant of either the wild-type or the cholesterol strain. Since the *K. phaffii* homolog for Say1 could not be identified yet, neither by annotation nor by specific BLAST searches, we were unable to conduct tests to ascertain whether the deletion of *SAY1* would result in the accumulation of acetylated sterol species. Nevertheless, our experiments revealed a noteworthy finding: substantial quantities of free sterols were excreted from both the *K. phaffii* and *S. cerevisiae* cholesterol strains, exceeding 200 µg/L cell supernatant (Figure 8A). Strikingly, the ratios of the respective excreted sterol species did not align with those found within the cells. Specifically, while the *K. phaffii* and *S. cerevisiae* cholesterol strains primarily accumulated 7-dehydrocholesterol and cholesterol, the cell supernatants were predominantly enriched in cholesta-5,7-24(25)-trienol (Figure 8A). *K. phaffii* cells also naturally secreted small amounts of ergosterol (up to 7 µg/L), which was not the case for *S. cerevisiae*. On the transcriptional level, several ATP-binding cassette (ABC) transporters were found to be upregulated in the cholesterol-producing *K. phaffii* strain, namely *SNQ2* (LFC 1.13), *PDR5* (LFC 0.71), and *PDR12* (LFC 0.78) (Figure 7B). *PDR15* (LFC −1.96), on the contrary, was found to be downregulated. In order to investigate the role of these transporters in sterol excretion, we deleted the respective genes in the wild-type and cholesterol strain and quantified the sterol species in the culture supernatants (Figure 8C). In the *K. phaffii* wild-type strain, the deletion of *SNQ2*, *PDR12*, and especially *PDR15* had the most significant influence on ergosterol excretion, while no severe impact on growth was observed. The deletion of *PDR15* lowered ergosterol excretion by 84%. Deletions of *SNQ2*, *PDR5*, *PDR12* and *PDR15* lowered the levels of excreted sterols by 40–60% in the cholesterol strain.

## 3. Discussion

Cells can adapt to different stresses by varying their lipid compositions. For example, yeasts can adapt to temperature changes by modulating their phospholipid compositions [43], fatty acid chain lengths and degrees of fatty acid saturation [46,47]. Due to the fact that sterols comprise ~30 mol% of the total PM lipids [48] and the structural rearrangements necessary to compensate the higher planarity of the cholesterol ring system in the membranes [31], it is not surprising that cholesterol production massively impacts diverse cellular processes in yeast, including the composition and structure of the membrane bilayer forming lipids. For example, in order to maintain cell membrane fluidity, yeast cells producing cholesterol seem to counterbalance the higher planar order of the sterol ring structure by producing more polyunsaturated fatty acids (Figure 3).

We also observed a strong impact on sphingolipid biosynthesis in the cholesterol strain. Sterols and sphingolipids are enriched in lipid rafts [49], where they contribute to diverse cellular processes. In vitro partitioning experiments between membranes revealed that sterols have an affinity for membranes with high content of sphingolipids [50], and the common hypothesis is that the two lipid species coevolved to provide optimal interaction properties between them [51]. Several studies have previously shown a link between sphingolipid and sterol profiles. For example, *erg* mutants show altered sphingolipid profiles [52], and genetic changes in sphingolipid biosynthesis can compensate for the lethality caused by *erg* mutants in *S. cerevisiae* [44]. We noticed the enrichment of IPC and decreased amounts of Cer, Hex-Cer and MIPC in the cholesterol strain. These changes could be explained by the respective up- and downregulation of the genes in the biosynthetic pathways, which suggests a global transcriptional response mechanism to the alterations in sterol biosynthesis.

Sterol biosynthesis is a very energy-intensive process. The synthesis of a single ergosterol molecule entails the sequential activity of 23 predominantly membrane-associated enzymes, orchestrating the conversion of 15 acetyl-CoA compounds to ergosterol [53]. This process also consumes 10 molecules of glucose [21], which is why excessive sterol production is highly uneconomic for a unicellular organism like yeast. In the cholesterol strain, we found highly upregulated sterol biosynthesis, which indicates that the main regulatory mechanisms that normally balance sterol homeostasis must be specifically controlled by the presence or absence of ergosterol. Our results demonstrate that sterol regulatory element binding proteins (SREBPs) such as Upc2, which acts as a sterol sensor and induces sterol biosynthetic genes upon sterol depletion [37], transcriptionally upregulated sterol biosynthetic genes in the cholesterol-producing strain (Figure 7C). Since Upc2 was reported to be unable to bind cholesterol [53], we conclude that Upc2 is not only upregulated but also constantly activated due to the absence of ergosterol. Despite the deletion of *UPC2*, the cholesterol levels and its precursors were not reduced to the wild-type sterol levels. This observation suggests the involvement of additional regulatory elements that remain active, thereby contributing to the overproduction of sterols in the cholesterol strain. Exploring these additional regulatory elements could present an intriguing avenue for a subsequent investigative study.

To date, barely any knowledge exists regarding if and how sterols may be exported from yeast cells. A *S. cerevisiae pdr5*∆ knockout strain was previously shown to be more sensitive to steroids [54]. Pdr5 belongs to the family of asymmetric ABC transporters, which is also the case for the human sterol transporters ABCG5/ABCG8. ABCG5/ABCG8 are also the body’s primary defense against the accumulation of neutral sterols (cholesterol and phytosterols) from the diet (reviewed by [55]). Our results demonstrate that a similar detoxification mechanism might exist in yeast with the help of diverse ABC transporters, and they indicate that these transporters potentially have different substrate specificities. While Pdr15 seems to be mainly responsible for the export of ergosterol, Snq2, Pdr5 and Pdr12 may be rather involved in the export of non-native sterols (Figure 8B,C).

In this study, we present a comprehensive multilevel approach aimed at investigating sterol-related dysfunctions and stress response mechanisms in yeast. Besides analyzing a human sterol-producing *K. phaffii* strain, we extended our exploration to a cholesterol-producing *S. cerevisiae* strain to validate and highlight key findings, revealing the conservation of crucial response mechanisms. For instance, our observations indicate that sterol esters and triacylglycerols (TAGs) accumulate in the cholesterol-producing *S. cerevisiae*, accompanied by an increase in lipid droplet sizes. Notably, cholesterol precursors are hyperexcreted from the cells, unveiling a significant and conserved detoxification mechanism. It is worth noting that all reported findings were more pronounced in *K. phaffii*.

For several reasons, *K. phaffii* is considered a better model organism than *S. cerevisiae* to study lipid-associated phenomena [56]. Firstly, *K. phaffii* did not undergo whole-genome duplication, enhancing the impact of genetic disturbances. Secondly, the phospholipids in *K. phaffii* generally display a lower degree of fatty acyl chain saturation, resembling those found in mammalian cells [57]. Lastly, the closer relation of *K. phaffii* to fungi such as *Schizosaccharomyces pombe*, an important model organism in eukaryotic cell biology, adds to its suitability [58]. The transcriptome data and lipid profiles presented in this publication not only offer novel insights but also provide an excellent foundation for diverse follow-up studies. This work contributes to the understanding of sterol-related processes and sets the stage for further exploration in yeast biology and beyond.

## 4. Materials and Methods 

### 4.1. Cloning and Strain Construction

PCR amplification was performed using Phusion™ DNA polymerase (Thermo Fisher Scientific Inc., St. Leon-Rot, Germany). Plasmids were cloned applying the Gibson assembly [25] and all constructs were verified by restriction analysis and DNA sequencing. For the HA-tagging of Pry2 and Flag-tagging of Epx1, CRISPR/Cas9 was applied [59]. Briefly, sgRNA targeting sequences of the pCas9-Hyg plasmids were adapted from Lehmayer et al., 2022 [60] to target *PRY2* for HA-tagging, *EPX1* for 3FLAG-His_6_-tagging and *UPC2*, *SNQ2*, *PDR5*, *PDR12* and *PRD15* for gene deletions (Appendix A). Repair and insertion plasmids were either constructed from pPpKC2 [61] and cut out using *Smi*I, or amplified by PCR. Bands of the correct size were gel-purified. The transformation of *K. phaffii* was performed according to the condensed protocol of Lin-Cereghino et al. [62]. For gene editing using CRISPR/Cas9, we followed the protocol of Weninger at al., 2016 [63]. In short, cells were co-transformed with 100 ng of pPpHyg-Cas9 plasmid and 500 ng of the respective repair cassette (Appendix A) and plated onto YPD containing 200 µg/mL hygromycin. Correct integration of cassettes into the yeast genome was verified by cPCR and sequencing. For gene knockout via CRISPR/Cas9 without a repair cassette, cells were transformed with 100 ng of pPpHyg-Cas9 targeting the respective gene (Appendix A). The correct integration of expression cassettes into the yeast genome was confirmed by colony PCR and sequencing.

### 4.2. Cultivation of Cells

In this study, the *K. phaffii* strain CBS7435 (NRRL Y-11430, ATCC 76273) *his4*∆ [64] was used as the wild-type strain for further engineering. All yeast strains used in this study (Appendix A) were grown on minimal dextrose (BMD) media (2% glucose, 13.4 g L^−1^ Yeast Nitrogen Base (without amino acids), 4 × 10^−5^% biotin and 0.4% histidine to permit growth of auxotrophs) [65]. Cultures were propagated at 28 °C in baffled flasks and analyzed either at the early (OD_600_~4) or middle exponential (OD_600_~10) growth phase, as indicated.

### 4.3. Transcriptome Analysis

RNA isolation was conducted by the SV Total RNA Isolation System (Promega Coorporation, Madison, WI, USA) as instructed by the provider’s manual. Briefly, cells were harvested during the early exponential phase. The disruption of cells was performed with glass beads by the alternate use of the vortex with max. speed and chilling on ice, for 30 s each, repeated 8 times. Pure RNA samples were immediately collected in 100 µL of RNase-free water and treated with liquid N_2_ prior to storage at −80 °C. The quality control of RNA isolates was conducted via denaturing gel electrophorese [66] and UV–Vis spectrophotometric measurements with the NanoDrop 2000 UV–Vis Spectrophotometer (Thermo Scientific, Waltham, MA, USA). Stranded library prep and sequencing was performed by Genewiz (Genewiz Azenta, Leipzig, Germany). The NEBNext^®^ Ultra™ II Directional RNA Kit (New England Biolabs Inc., Ipswich, Massachusetts, USA) was used for library preparation, and fragments were sequenced on an Illumina machine to generate 150 bp paired-end reads. Quality controls of the isolated RNA and the fragment library were performed according to Genewiz standards. Sequencing data have been deposited to the ENA under PREJB70602. The reads were trimmed using Trimmomatic v0.39 [67] and were subsequently mapped onto the *K. phaffii* CBS7435 reference assembly [68,69] using the STAR aligner v2.7.5a [70]. The quality of the raw reads and the mapping were assessed using FastQC v0.11.9 [71] and Qualimap v2.2.1 [72], respectively. Differential expression analysis based on read counts calculated by STAR was performed in R [73]. Log fold changes of gene expression were determined with edgeR v3.30.3 [74] using a Fisher’s exact test on normalized read counts based on the robust estimation of dispersion [75]. Principal component analysis was performed using the package PCAtools v2.0.0 [76]. The volcano plot was generated using the R package EnhancedVolcano v 3.17 (https://bioconductor.org/packages/release/bioc/html/EnhancedVolcano.html, accessed on 1 August 2023).

### 4.4. SDS-PAGE and Immunoblot Analysis

Here, 4 OD_600_ units of exponentially growing cells or aliquots of culture supernatants were harvested by brief centrifugation and stored at −80 °C. Cell pellets were thawed on ice and lysed in 300 µL of lysis solution (1.85 M NaOH; 7.5% of ß-mercaptoethanol). Proteins were precipitated by the addition of 300 µL of 50% ice-cold trichloroacetic acid (TCA) for an hour on ice, or overnight at 4 °C. Afterwards, proteins were pelleted by centrifugation and the pellets washed twice with ice-cold water. Precipitated proteins were solubilized in 50 µL of NuPAGE™ sample buffer (Thermo Fisher Scientific Inc., Germany), supplemented with 4% β-mercaptoethanol and 30% 1 M Tris and heated to 75 °C for 15 min. Then, 10 µL of sample was resolved on 12% Bis-Tris NuPAGE™ gels. Separated proteins were transferred electrophoretically onto a nitrocellulose membrane, which was then blocked for an hour with TBST (30.3 g·L^−1^ Tris; 87.6 g·L^−1^ NaCl; 0.003% Tween20) supplemented with 5% bovine serum albumin (BSA). Washing steps were conducted 3 times for 10 min with TBST. Proteins on the membranes were probed overnight at 4 °C by the addition of the appropriate primary antibody: mouse ANTI-FLAG^®^ M2-Peroxidase (HRP) antibody (1:4000; Sigma Aldrich, Vienna, Austria); peroxidase-conjugated anti-HA 3F10 from rat (1:2500; Roche); rabbit anti-GAPDH (1:5000; Institute of Biochemistry, Graz University of Technology, Graz, Austria) [25]; and rabbit anti-Pma1 (1:1000; Institute of Biochemistry, Graz University of Technology, Austria) [25]. After washing three times with ≥10 mL with TBST, membranes were either used directly for immunodetection or incubated with an appropriate HRP-conjugated secondary antibody—goat anti-rabbit IgG–peroxidase antibody A9169 (1:10,000, Sigma Aldrich, Vienna, Austria)—and then washed three times with TBST as above. Enhanced chemiluminescent signal detection (Clarity Max Western ECL Substrate, Bio-Rad, Vienna, Austria) was used to visualize immunoreactive bands. The PageRuler™ pre-stained protein ladder (Thermo Scientific™) was used as a molecular weight marker.

### 4.5. Plasma Membrane Isolation

The isolation of the plasma membrane (PM) was performed as described by Grillitsch et al., 2014 [57], with slight modifications. The growth conditions as described above were scaled up to gain wet weights ranging from 44 to 77 mg cell pellet. Cells were harvested during the middle exponential phase. Crude PM was dissolved in 10 mM Tris HCl, pH 7.4, and transferred to Eppendorf 1.5 mL vessels. PM samples were pelleted and stored at −80 °C for HPLC/GC or quality control analyses. Protein concentrations were determined by the Pierce BCA Protein Assay Kit™ as described in the provider’s manual (Thermo Fisher Scientific).

### 4.6. Lipid Analysis

#### 4.6.1. Sterol Analysis via GC-MS

The procedure for intracellular sterol quantification was performed according to the description of [77], with minor modifications. Cell pellets corresponding to 15 OD_600_ units were harvested and dispersed in 1 mL of 0.2% pyrogallol in methanol. After the addition of 0.4 mL of 60% aqueous KOH, the internal standard, cholesterol-d7 (Avanti Polar Lipids, Alabaster, AL, USA), was added in total amounts of 10 µg. Then, samples were incubated for 2 h at 90 °C in a sand bath. Extraction was performed twice according to protocol [77]. For the analysis of excreted sterols, cells were pelleted at the late exponential phase, and 3 mL of cell supernatant was transferred to Pyrex^®^ glass tubes. Then, 10 µg of cholesterol-d7 was added and sterols extracted by the addition of 1.5 mL heptane and vigorous mixing for 25 min. Pyrex tubes were then centrifuged at 4000 rpm for 20 min. Extractions were performed twice and heptane phases transferred to fresh tubes. N_2_-dried extracts of intracellular and excreted sterols were then derivatized with 10 µL of pyridine and 50 µL of N,O-bis (trimethylsilyl)trifluoroacetamide for 25 min under moderate shaking. The samples were solved in 200 µL of ethyl acetate and analyzed on a gas chromatography system, the Hewlett-Packard 6890 GC combined with a Hewlett Packard 5973 mass selective detector.

#### 4.6.2. Thin Layer Chromatography

For thin layer chromatography, 60 OD_600_ units of exponentially growing yeast cells (OD_600_ of 4–6) were pelleted by centrifugation. Cell disruption was performed by the addition of CHCl_3_/MeOH (2:1; *v*:*v*) and glass beads, under shaking with 1500 rpm on a VXR basic Vibrax™ for 1 h. After brief centrifugation, the supernatant was transferred to a fresh Pyrex^®^ glass tube. Next, 2 mL of 0.0034% of MgCl_2_ was added; the mixtures were shaken with 1500 rpm for 5 min and subjected to centrifugation at 2500 rpm for 5 min. The upper phase was discarded. Then, 2 mL of a 2 M KCl/MeOH (1:1; *v*:*v*) mixture was added, shaken with 1500 rpm for 5 min and subjected to centrifugation at 2500 rpm for 5 min. The upper phase was discarded. Washing was performed with 1.5 mL of CHCl_3_/MeOH/H_2_O (0.03:4.8:4.7; *v*:*v*:*v*), and, after mixing and centrifugation, the top phase and interphase were discarded. Lipid extracts in the lower phase were dried under a nitrogen stream and dissolved in 100 µL of CHCl_3_/MeOH (2:1; *v*:*v*). Then, 5 µL of the sample as well as 3 µL of lipid standard (2 mg mL^−1^) were applied onto TLC-Silica plates with a Hamilton syringe. As the first running solvent, a mixture of petroleum ether/diethylether (1:1, per vol.) was used for the separation of lipid species, and the second running solvent consisted of petroleum ether/diethylether/acetic acid (19:1:0.02, per vol.). After separation, the TLC plates were briefly dipped into the staining solution (0.3 g MnCl; 60 mL H_2_O; 60 mL MeOH; 4 mL H_2_SO_4_) and lipid bands were charred for 20 min at 120 °C. Quantifications of band intensities were performed using Fiji [36].

#### 4.6.3. HPLC-MS Analysis

All solvents were obtained as HPLC-MS grade. Water, 2-propanol and phosphoric acid were purchased from Roth (Karlsruhe, Germany), methanol from J. T. Baker (Austin, TX, USA), formic acid from Sigma (Vienna, Austria) and ammonium acetate was purchased from Merck (Darmstadt, Germany).

Samples were dissolved in 300 µL of isopropanol/methanol/water (30:15:5, *v*:*v*:v). Chromatographic separation was performed using a 1290-UHPLC system (Agilent, Waldbronn, Germany) equipped with an BEH-C18-column, 2.1 × 150 mm, 1.7 μm (Waters, Manchester, UK). The autosampler compartment was set to 8 °C and 4 µL samples were injected. A binary gradient was applied. Solvent A was water; solvent B was 2-propanol. Both solvents contained phosphoric acid (8 μM), ammonium acetate (10 mM) and formic acid (0.1 vol%). The linear gradient started at 40% solvent B at a constant flow rate of 0.2 mL/min and increased to 100% solvent B within 20 min. In the following 4 min, the solvent B percentage was kept at 100%. The column was re-equilibrated for 5 min, resulting in a total HPLC run time of 30 min. The column compartment was kept at 50 °C. A 4670 triple quadrupole mass spectrometer (Agilent) equipped with an ESI source was used for analysis. The following source parameters were used: source temperature: 300 °C, sheath gas (N_2_) temperature: 400 °C. The capillary voltage was 3.5 kV in positive ionization mode. Samples were analyzed in MRM mode and data analysis was performed using the MassHunter 10.0 software package (Agilent).

#### 4.6.4. Fluorescence Microscopy

##### LD Labeling and Microscopy

LD were labeled using BODIPY 493/503 for 15 min at RT. Then, 3D stacks were acquired using a LEICA SP8 confocal laser scanning microscope with spectral detection (Leica Microsystems, Inc.), an HCX PL APO 63x NA 1.4 oil immersion objective, a constant photomultiplier gain and 44 × 44 × 300 nm (x/y/z) sampling. BODIPY 493/503 was excited at 488 nm and emission was detected between 500 and 550 nm. Fluorescence and transmission images were acquired simultaneously. The brightness and contrast of fluorescence data were adapted for visualization.

##### Image-Based Quantification of LD

The deconvolution of 3D fluorescence data was performed using Huygens professional (SVI, Inc., Pathumthani, Thailand), a theoretical point-spread function and the classic maximum-likelihood estimation algorithm (SNR 7, 5 iterations). In transmission images, diffracted light manifests as halos of alternating black and white bands, whose thickness varies with the axial focus position. The midst of the cells was identified as the optical section within the 3D data set where the thickness of the white halo ring was at its minimum. Individual cell areas were manually registered using selection tools in the open-source software Fiji (version 2.9.0) [36]. Volume quantification of fluorescently labeled LD was performed using the Fiji 3D suite plugin (version 1.7.0) [78]. LD were segmented from 3D fluorescence data sets by thresholding (same value for all stacks). The total LD volume/total cell area was computed.

### 4.7. Acetyl-CoA and Malonyl-CoA Quantification

*K. phaffii* cell pellets were resuspended in 300 µL of MeOH/ACN/H_2_O (40:40:20; *v*:*v*:*v*) and the suspension was mixed with an equal volume of 0.25–0.5 mm glass beads (Roth A533.1). Cells were lysed employing a Bead Mill Max (VWR) at 6 m/s setting for 30 s. The mixture was centrifuged for 1 min at 20.000*g* × and the extract was removed from the glass beads. The precipitated protein was resolubilized in 1% SDS at 95 °C for 5 min and the total protein content was determined employing the Pierce BCA Protein Assay Kit for the normalization of samples. The extract was centrifuged at 20.000*g* × for 5 min to remove residual particles and then subjected to HILIC-MS employing a ZICpHILIC column (5 µm, 150 × 2.1 mm), with solvent A being 20 mM ammonium formate at pH = 9.6 and solvent B being ACN. The following gradient was employed: 0–2 min 10% A, 2–14 min linear increase to 40% A, 14–20 min linear increase to 95% A, 20–23 min 95% A followed by re-equilibration at 10% A for 7 min. The MS system used was a timsTOF Pro (Bruker Daltonics, Bremen, Germany) operated with a VIP HESI source set to 200 °C in the default 4D-Lipidomics method. The identity of analytes was confirmed by employing authentic standards (Sigma Aldrich) by matching retention times, collisional cross-sections and exact mass and fragmentation patterns. 

## Figures and Tables

**Figure 1 ijms-25-00781-f001:**
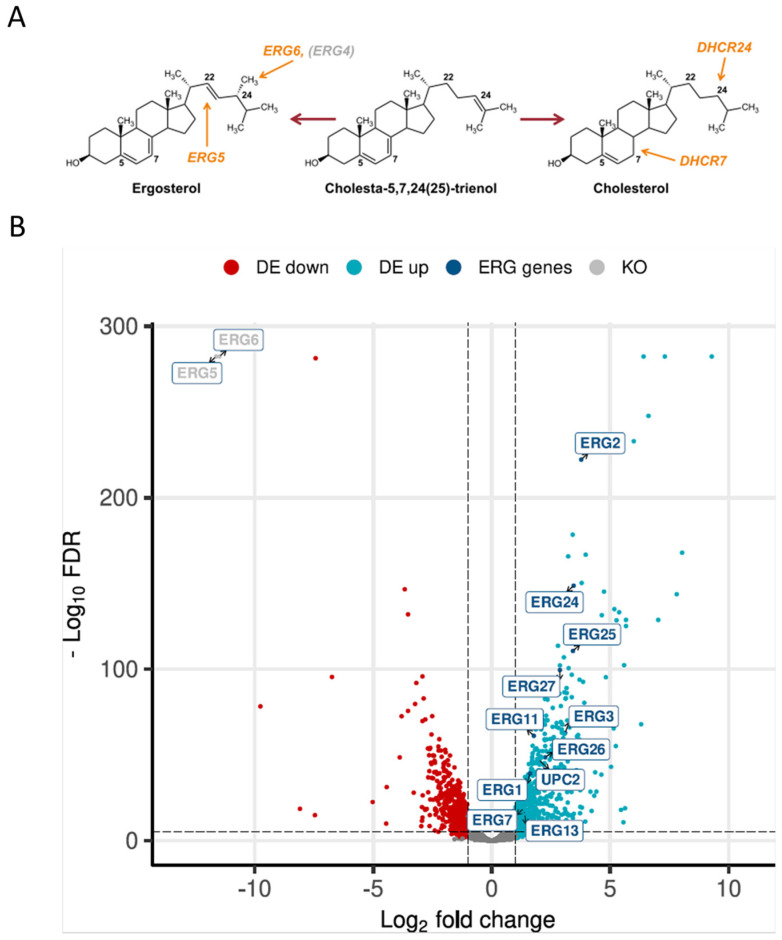
Production of cholesterol severely affects gene expression in *K. phaffii*. (**A**) Cholesterol production was achieved by the deletion of *ERG5* and *ERG6*, while genes for DHCR7 and DHCR24 were inserted [15]. (**B**) Volcano plot of RNA-seq data was generated using the R package EnhancedVolcano v 3.17. Downregulated genes are highlighted in red, upregulated genes in light blue. Genes found to be upregulated in the ergosterol biosynthesis pathway are highlighted in dark blue. DE, differentially expressed. KO, knocked out genes.

**Figure 2 ijms-25-00781-f002:**
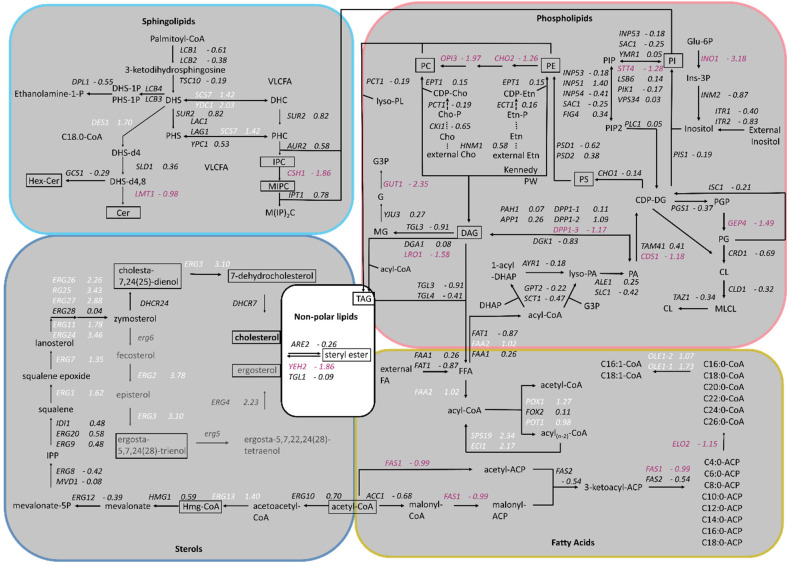
Changes in expression of lipid biosynthesis genes in cholesterol-producing *K. phaffii*. Genes of the cholesterol strain were compared to the parental strain. Lipid species analyzed in the study are boxed. Fold changes of genes are indicated by color: white, upregulated genes; red, downregulated genes; black, no statistically significant changes. (FDR < 0.05, pathways taken from Adelantado et al., 2017 [24]).

**Figure 3 ijms-25-00781-f003:**
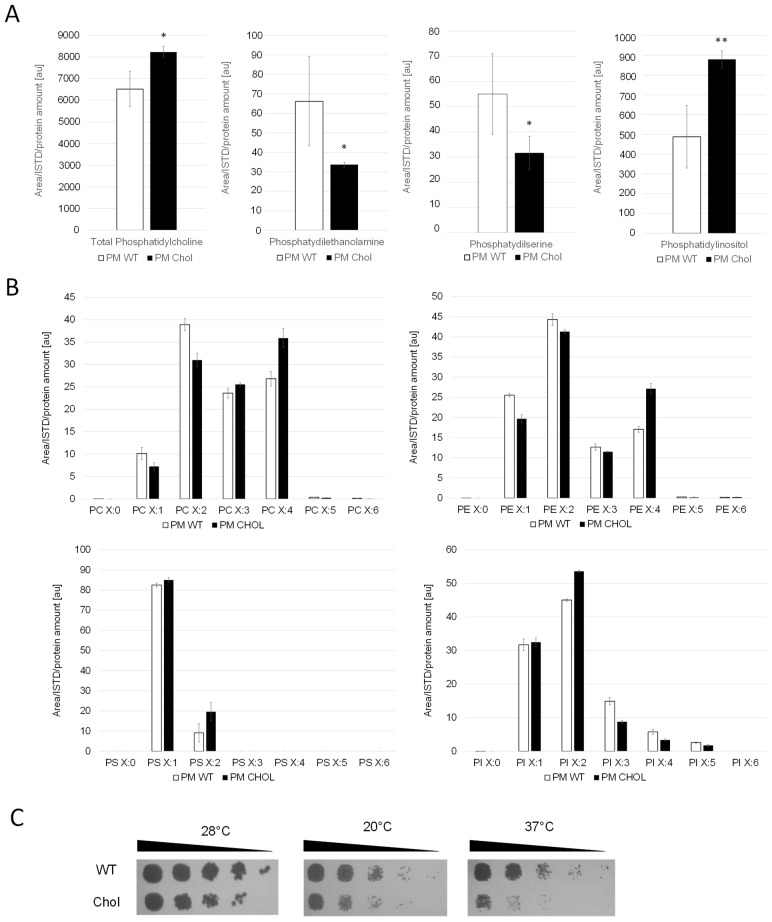
Cholesterol production influences phospholipid levels and saturation of their fatty acid tails. (**A**) Plasma membrane (PM) fractions of the wild-type (CBS7435 *his4*Δ) and cholesterol-producing *K. phaffii* strain (yMH468) were generated and analyzed for phospholipid content by LC-MS as described in Materials and Methods. Mean values and standard deviations of biological triplicate measurements are shown. * *p* value < 0.01, ** *p* value < 0.0001, determined by two-tailed Student’s *t* test. *p* values were obtained for cholesterol compared to the WT strain. (**B**) In addition to total amounts, different subspecies were analyzed, quantified and levels compared to wild-type strains. (**C**) The same strains as under (**A**) were cultivated as described in Materials and Methods, and then samples of a set of 5-fold serial dilutions were spotted using a multipronged inoculator on an agar plate containing YPD, and, after incubation for 72 h at 28 °C, 20 °C or 37 °C, the resulting growth was recorded.

**Figure 4 ijms-25-00781-f004:**
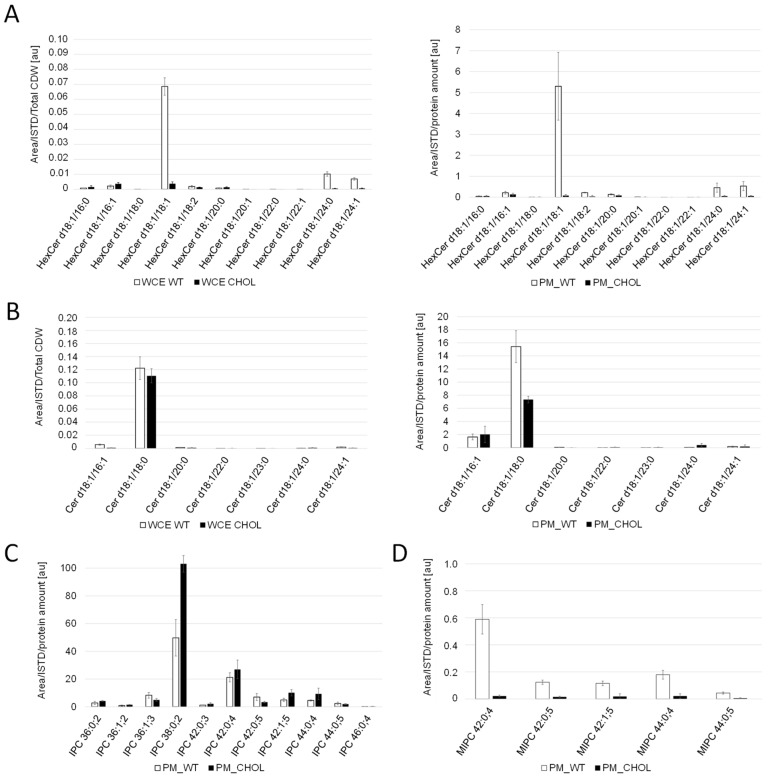
Sphingolipid analysis of cholesterol and wild-type *K. phaffii*. Whole cell extracts (WCE) and purified plasma membrane (PM) fractions of the wild-type and cholesterol strains from middle exponential phase were used for LC-MS analysis of (**A**) hexosylceramide (Hex-Cer) and (**B**) ceramide (Cer). For analysis of (**C**) inositol phosphorylceramide (IPC) and (**D**) mannosylinositol phosphorylceramide (MIPC), only data obtained for plasma membrane fractions are shown, since levels in whole cell extracts were too low for proper analysis. Sphingolipid molecular species are expressed as “Sphingolipid-class (Long-chain-base/Fatty-acyl)”. Long chain bases and fatty acyls are expressed as XX:YY;Z (XX: number of carbons; YY: number of C-C double bonds, Z: number of hydroxyl groups). Mean values and standard deviations of biological triplicates are shown.

**Figure 5 ijms-25-00781-f005:**
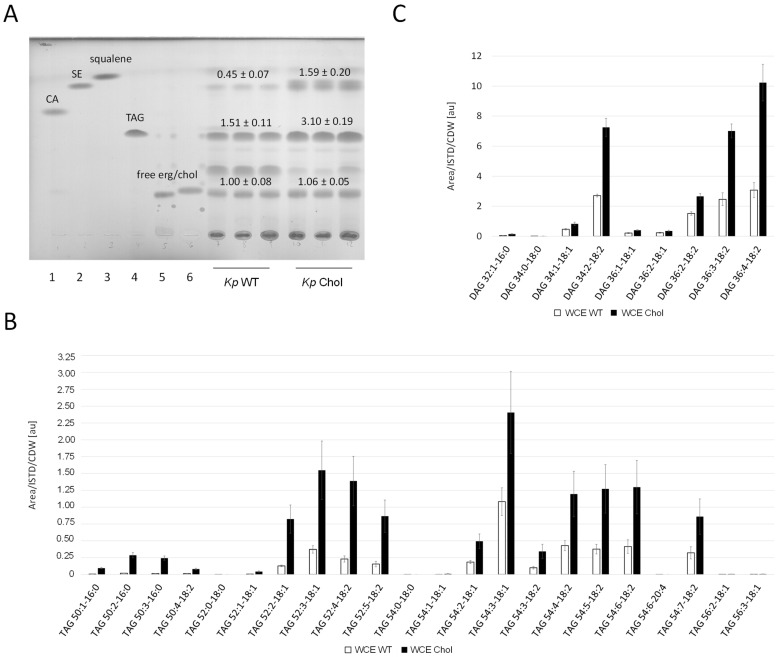
Neutral lipid analysis in wild-type and cholesterol strains. (**A**) *K. phaffii* and *S. cerevisiae* wild-type and cholesterol strains were cultivated at 28 °C for 24 h until they reached early exponential phase, harvested by centrifugation, and analyzed by thin layer chromatography. Applied standards were 6 µg of (1) cholesteryl acetate (CA), (2) cholesteryloleate/sterol ester (SE), (3) squalene, (4) triolein (TAG (triacylglycerol)), (5) free ergosterol, and (6) free cholesterol. Band intensities obtained on TLCs were quantified using Fiji [36] and signal quantifications are presented as mean +/− standard deviations from three biological replicates. (**B**) TAG and (**C**) diacylglycerol (DAG) analysis was performed with *K. phaffii* wild-type and cholesterol strains applying LC-MS. Cells were cultivated as usual at 28 °C and harvested when they reached early exponential phase. 300 OD_600_ units were used directly for lipid extraction (whole cell extracts, WCE). LC-MS analysis was performed as described under Materials and Methods. TAG and DAG species are presented as [number of acyl carbons: number of double bonds—fatty acid: number of double bonds]. Mean +/− standard deviations from three biological replicates are shown.

**Figure 6 ijms-25-00781-f006:**
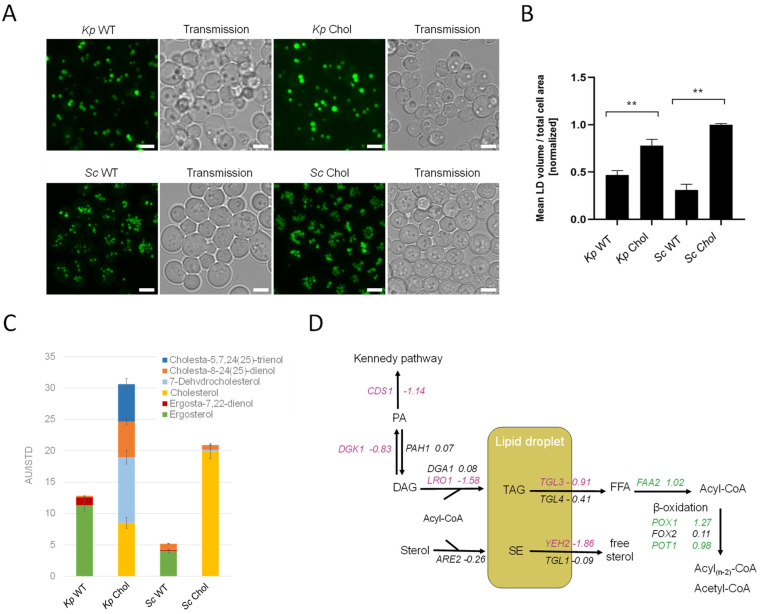
Lipids accumulate in cholesterol-producing yeast strains. (**A**) *K. phaffii* and *S. cerevisiae* wild-type and cholesterol strains were cultivated at 28 °C for 24 h until they reached early exponential phase and harvested by centrifugation. Lipid droplets were labeled using BODIPY 493/503 for 15 min at RT and observed under the microscope. Bar = 5 µm. (**B**) Data obtained from images shown under (**A**) were analyzed and quantified as described in Materials and Methods. Mean values and standard deviations of signals obtained for >100 cells are shown. ** *p* value < 0.0001, determined by two-tailed Student’s *t* test. *p* values were obtained for cholesterol compared to the WT strain. (**C**) The same strains as in (**A**) were cultivated at 28 °C for 24 h until they reached early exponential phase, and total sterols were extracted and analyzed by GC-MS as described in Materials and Methods. Data are presented as mean from three biological replicates. (**D**) Changes in expression of lipid droplet metabolic genes in cholesterol-producing *K. phaffii*. Genes of the cholesterol strain were compared to wild type. Fold changes of genes are indicated by color: green, upregulated genes; red, downregulated genes; black, no statistically significant changes (FDR < 0.05).

**Figure 7 ijms-25-00781-f007:**
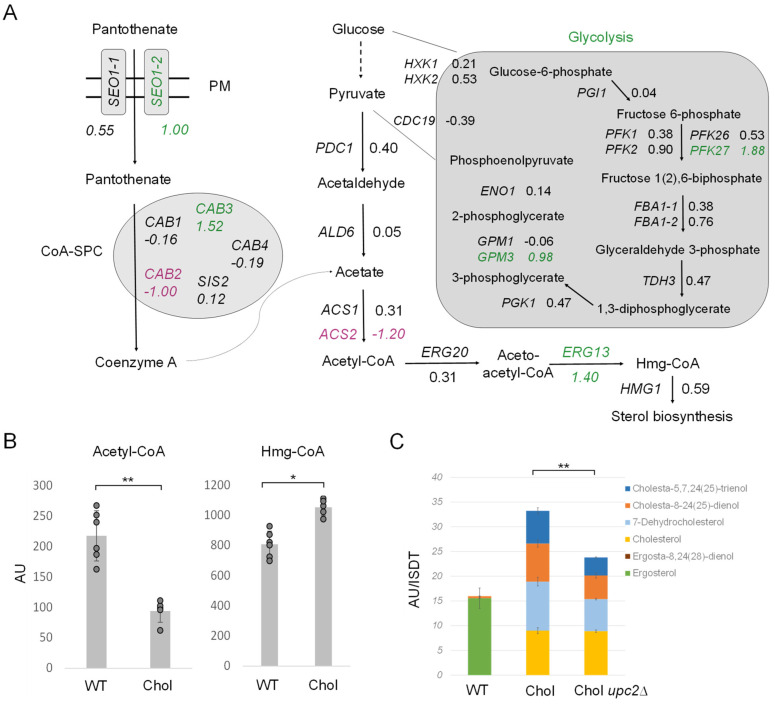
Regulation of acetyl-CoA biosynthesis and the early mevalonate pathway in cholesterol-producing *K. phaffii* cells. (**A**) Changes in expression of genes involved in acetyl-CoA biosynthesis in cholesterol-producing *K. phaffii*. Genes of the cholesterol strain were compared to wild type. Fold changes of genes are indicated by color: green, upregulated genes; magenta, downregulated genes; black, no significant changes (FDR < 0.05). (**B**) *K. phaffii* wild-type and cholesterol strains were cultivated at 28 °C for 24 h until they reached early exponential phase, harvested by centrifugation and acetyl-CoA, and Hmg-CoA levels determined as described in Materials and Methods. Data are presented as mean from six biological replicates. AU, arbitrary units. Error bars, standard deviations, ** *p* value < 0.0001, * *p* value < 0.01 determined by two-tailed Student’s *t* test. *p* values were obtained for the cholesterol strain compared to the wild-type strain, respectively. (**C**) *K. phaffii* wild-type and cholesterol strains, as well as a cholesterol *upc2*∆ strain (yAR097), were cultivated and harvested as described above and subjected to sterol extraction and analysis by GC-MS in biological triplicates.

**Figure 8 ijms-25-00781-f008:**
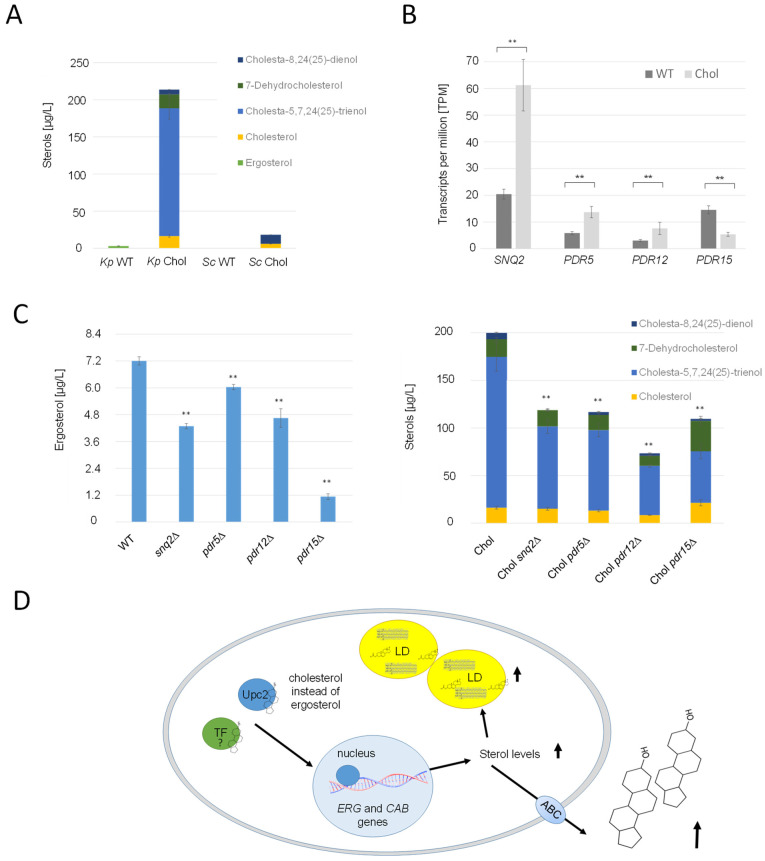
Overproduced sterols are excreted from *K. phaffii* by ABC transporters. (**A**) *K. phaffii* and *S. cerevisiae* wild-type and cholesterol strains were cultivated at 28 °C for 24 h until they reached early exponential phase and cells separated from culture supernatants by centrifugation. Sterols were extracted from culture supernatants and analyzed for sterol content as described in Materials and Methods. (**B**) Comparison of differential regulation of ABC transporter genes *SNQ2*, *PDR5*, *PDR12* and *PDR15* in the wild-type and cholesterol strain. Transcripts per million (TPM, fragment counts normalized for gene length) are presented as mean +/− standard deviations from three biological replicates. ** *p* value < 0.0001, determined by two-tailed Student’s *t* test. *p* values were obtained for cholesterol compared to the WT strain. (**C**) The *K. phaffii* wild-type strain, and otherwise isogenic derivatives carrying deletions of *snq2*∆ (yAR070), *pdr5*∆ (yAR056), *pdr12*∆ (yAR077) and *pdr15*∆ (yAR093), as well as the *K. phaffii* cholesterol strain, and otherwise isogenic derivatives carrying deletions of *snq2*∆ (yAR074), *pdr5*∆ (yAR053), *pdr12*∆ (yAR077) and *pdr15*∆ (yAR092), were cultivated at 28 °C for 24 h until they reached early exponential phase, and cells were separated from culture supernatants by centrifugation. Sterols were extracted from culture supernatants and analyzed for sterol content as described in Materials and Methods. Data are presented as mean from three biological replicates. Error bars, standard deviations, ** *p* value < 0.0001, determined by two-tailed Student’s *t* test. *p* values were obtained for the knockout strains compared to the respective wild-type strain. (**D**) Primary mechanisms observed to be differentially regulated in the cholesterol strain. The sterol-responsive element Upc2, while incapable of binding cholesterol, serves as the trigger for the transcription of various *ERG* genes, and potentially *CAB* genes as well. Excessive stimulation of ergosterol biosynthesis results in the onset of lipotoxicity. To counteract the adverse effects of lipotoxicity, cells respond by sequestering surplus sterols within lipid droplets (LD) or excreting them into the culture supernatant with the aid of ABC transporters.

## Data Availability

All sequenced reads are accessible on EMBL-EBI under accession number PRJEB70602 (https://www.ebi.ac.uk/ena/browser/view/PRJEB70602, accessed on 6 December 2023). Other original data presented in this publication can be obtained from the lead contact upon request.

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
