# Peer review of "Human Sterols Are Overproduced, Stored and Excreted in Yeasts"

_ijms, 2024, doi:10.3390/ijms25020781_

Round 1

Reviewer 1 Report

Comments and Suggestions for Authors

In this manuscript, Radkohl and co-authors present a comprehensive study on yeast strains that produce cholesterol instead of ergosterol by investigating the cellular response mechanisms using transcriptome and lipidome approaches in the model Komagataella phaffii (syn Pichia pastoris) and Saccharomyces cerevisiae. The authors show that cholesterol and its precursors ae excreted into the culture supernatant, most likely by the action of ABC transporters Snq2, Pdr12 and Pdr15.

The study is interesting, presenting clear and sound data. The manuscript is very well written, in a logical and coherent way. Nevertheless, there are some issues that must be considered before the manuscript can be recommended for publication in IJMS.

-        Figure 5A is unclear; the significance of each band needs to be explained. For instance, it is not explained what the lowest bands represent. Also, one may guess which band is for free ergosterol/cholesterol, but the significance of the upper bands is not evident.

-        Figure 5 B, C: the significance of X axis points is confusing. Please explain in the figure legend the meaning of both abbreviations (i.e. DAG, TAG) and the corresponding number codes. under such circumstances, it is not clear which data are for Komagataella phaffii and which are for Saccharomyces cerevisiae.

-        The terms „lipidomics”, „lipidome” may be a bit far-fetched. Rather, the analysis of cholesterol/ergosterol or total lipid profiling is performed.

Author Response

In this manuscript, Radkohl and co-authors present a comprehensive study on yeast strains that produce cholesterol instead of ergosterol by investigating the cellular response mechanisms using transcriptome and lipidome approaches in the model Komagataella phaffii (syn Pichia pastoris) and Saccharomyces cerevisiae. The authors show that cholesterol and its precursors ae excreted into the culture supernatant, most likely by the action of ABC transporters Snq2, Pdr12 and Pdr15.

The study is interesting, presenting clear and sound data. The manuscript is very well written, in a logical and coherent way. Nevertheless, there are some issues that must be considered before the manuscript can be recommended for publication in IJMS.

Concern raised: Figure 5A is unclear; the significance of each band needs to be explained. For instance, it is not explained what the lowest bands represent. Also, one may guess which band is for free ergosterol/cholesterol, but the significance of the upper bands is not evident.

Response: Thank you for this comment. It is correct that a standard TLC only allows a rough discrimination of the different lipid species by comparing sizes with the respective lipid standards. We did not focus on the exact determination of the lowest and upper bands, since this would have required a more complex (GC or LC) analysis, which was not in the focus of this study. For clarity, we added an explanation in the manuscript (p 14).

Concern raised: Figure 5 B, C: the significance of X axis points is confusing. Please explain in the figure legend the meaning of both abbreviations (i.e. DAG, TAG) and the corresponding number codes. under such circumstances, it is not clear which data are for Komagataella phaffii and which are for Saccharomyces cerevisiae.

Response: A more concise explanation of the X-axis nomenclature and the meaning of abbreviations was added to the Figure legend. Also, the legend now points out that these data were derived using K. phaffii strains.

Concern raised: The terms „lipidomics”, „lipidome” may be a bit far-fetched. Rather, the analysis of cholesterol/ergosterol or total lipid profiling is performed.

Response: Point taken. We exchanged the term ‘lipidome’ for ‘total lipid profiling’ at diverse sites throughout the whole manuscript.

Reviewer 2 Report

Comments and Suggestions for Authors

The authors present a comprehensive analysis of mammalian sterol expressing K. phaffii strain on transcriptome and lipidome level. The exchange of ergosterol for cholesterol caused a downregulation of PS and PE biosynthesis, and upregulated PI and PC biosynthesis, with a concomitant shift towards polyunsaturated fatty acids.

 They find that sterol esters and triacylglycerols accumulate in such strains in enlarged lipid droplets, and interestingly, cholesterol precursors are potently excreted from the cells, which presents a mechanism for detoxification (these observations are also extended to S. cerevisiae strain in this work). They also indicate that the mechanism of sterol excretion in yeast appears to be based on diverse ABC transporters, which have different substrate specificities.

The repertoire of methods used is very diverse, which strongly supports the integrative approach used. A large amount of data is presented, nevertheless the inference is clear and logical, as the findings of the study are explained with accompanying references to relevant biochemical pathways. Data presentation in the Figures is well organized (except at few points, please see comments below) and Methods very concisely described.

Please find below a list of remarks which I hope you will find helpful.

Title: does not describe the contents well, maybe “Cellular response of yeast expressing mammalian sterols on transcriptome and lipidome level”, or similar?

Abstract: similarly to mammalian cells

Page 2: „have low normal intelligence” – this sentence should be reworded into a more correct language

Page 2: Komagatella phaffii

Page 2: “In this study, we show that the ABC transporters Pdr12 and Pdr15…” – this sentence would be more suitable for the conclusion of this article.

Page 3: “fold changes of genes assigned” – fold changes of gene expression assigned

Figure 1 byline: Definition of the abbreviation KO (probably knock-out?) missing

Figure 2: I propose to change the text font in the Figure for larger bold letters, at the moment it is poorly legible

Page 6: “Whole cell extracts and purified plasma membrane fractions (Supplementary Figure S1)” – propose to add “quality control in Supplementary Figure S1”

Page 6: “PA to CDP-DG” – PA (phosphatidic acid) abbreviation is not defined

Page 6: “PI to PIP by STT4” – STT4-encoded PI-kinase

Figure 3: axis labels could be without decimal points (and should not be commas), font should be increased to improve legibility

Page 8: “In contrast to S. cerevisiae…” – please explain, this comparison is not obvious from Figure 3b

Page 8: In S. cerevisiae, two ELO genes, ELO2 and its paralog ELO1,… - reference citations in the following two sentences are missing.

Page 8: Figure 4B is not cited in the text.

Figure 4A: axis labels: decimal points, not commas; folns should be increased

Figure 5: Figure legend should not include the comments on S. cerevisiae if those results are presented in the Supplementary Figure.

Page 10: legend to Figure 5: “Cells were cultivated as usually”

Page 13: “were transcriptionally upregulated in cholesterol producing K. phaffii”

Figure 7B: Panels are overlapping

Figure 7C: I propose not to use different shades of blue as possibly the distinction is difficult (also in 6C, 8A, 8C)

Figure 8C: structural graphics are poorly visible

Page 19: UV-VIS spectrophotometrical

Please check the following references for completeness: 16, 60, 70, 76, 78

Supplementary Figure S3: axis labels: decimal point not comma

Supplementary Figure S4: Figure legend with different font sizes. Molecular weight marker lanes should be visible (and also, non-manipulated western blots should be provided, even if for review purposes only).

Author Response

Reviewer 2

The authors present a comprehensive analysis of mammalian sterol expressing K. phaffii strain on transcriptome and lipidome level. The exchange of ergosterol for cholesterol caused a downregulation of PS and PE biosynthesis, and upregulated PI and PC biosynthesis, with a concomitant shift towards polyunsaturated fatty acids.

 They find that sterol esters and triacylglycerols accumulate in such strains in enlarged lipid droplets, and interestingly, cholesterol precursors are potently excreted from the cells, which presents a mechanism for detoxification (these observations are also extended to S. cerevisiae strain in this work). They also indicate that the mechanism of sterol excretion in yeast appears to be based on diverse ABC transporters, which have different substrate specificities.

The repertoire of methods used is very diverse, which strongly supports the integrative approach used. A large amount of data is presented, nevertheless the inference is clear and logical, as the findings of the study are explained with accompanying references to relevant biochemical pathways. Data presentation in the Figures is well organized (except at few points, please see comments below) and Methods very concisely described.

Please find below a list of remarks which I hope you will find helpful.

Title: does not describe the contents well, maybe “Cellular response of yeast expressing mammalian sterols on transcriptome and lipidome level”, or similar?

Response: We agree that finding a suitable title for such a complex study is not easy, and we gave this matter a lot of thought. However, the brief summary provided by Reviewer 2 above contains all of the points mentioned in our current title, which is why we still feel that it fits quite well.

Abstract: similarly to mammalian cells

Response: Correct. We fixed the mistake. 

Page 2: „have low normal intelligence” – this sentence should be reworded into a more correct language

Response: We adapted the sentence.

Page 2: Komagatella phaffii

Response: It is indeed called Komagataella phaffii.

Page 2: “In this study, we show that the ABC transporters Pdr12 and Pdr15…” – this sentence would be more suitable for the conclusion of this article.

Response: In order to already indicate a possible role of ABC transporters in sterol excretion (which was not published for yeast before), we would like to briefly mention this in the introduction. However, in order to not provide any major results, we deleted ‘Pdr12 and Pdr15’ from the text.

Page 3: “fold changes of genes assigned” – fold changes of gene expression assigned

Response: Good point. We changed this.

Figure 1 byline: Definition of the abbreviation KO (probably knock-out?) missing

Response: True. We added a definition to the Figure legend.

Figure 2: I propose to change the text font in the Figure for larger bold letters, at the moment it is poorly legible

Response: We played around with the font size, but using bold letters turned out to be counterproductive for visual perception, since single letters became even harder to read. In the end, such ‘large’ Figures greatly profit from online availability, since this allows the readers to open them separately and take a very close look at the different pathways.

Page 6: “Whole cell extracts and purified plasma membrane fractions (Supplementary Figure S1)” – propose to add “quality control in Supplementary Figure S1”

Response: We modified the text accordingly.

Page 6: “PA to CDP-DG” – PA (phosphatidic acid) abbreviation is not defined

Response: Definition of abbreviation was added.

Page 6: “PI to PIP by STT4” – STT4-encoded PI-kinase

Response: Correct. We modified the text accordingly.

Figure 3: axis labels could be without decimal points (and should not be commas), font should be increased to improve legibility

Response: We adapted Figure 3 accordingly.

Page 8: “In contrast to S. cerevisiae…” – please explain, this comparison is not obvious from Figure 3b

Response: The respective paragraph was rephrased to improve clarity.

Page 8: In S. cerevisiae, two ELO genes, ELO2 and its paralog ELO1,… - reference citations in the following two sentences are missing.

Response: Respective references 35 and 36 were added to the manuscript.

Page 8: Figure 4B is not cited in the text.

Response: Citation was added on page 11.

Figure 4A: axis labels: decimal points, not commas; folns should be increased

Response: Changes were made as suggested.

Figure 5: Figure legend should not include the comments on S. cerevisiae if those results are presented in the Supplementary Figure.

Response: Correct. S. cerevisiae was eliminated from the Figure description.

Page 10: legend to Figure 5: “Cells were cultivated as usually”

Response: Text was adapted.

Page 13: “were transcriptionally upregulated in cholesterol producing K. phaffii”

Response: ‘cholesterol producing’ was added to the sentence.

Figure 7B: Panels are overlapping

Response: Thank you. Panels were separated.

Figure 7C: I propose not to use different shades of blue as possibly the distinction is difficult (also in 6C, 8A, 8C)

Response: We changed the color for 7-DHC to light blue, and the two colors should now be easily distinguishable.

Figure 8C: structural graphics are poorly visible

Response: Sizes of structural graphics were maximized and visibility should be improved now.

Page 19: UV-VIS spectrophotometrical

Response: Adapted.

Please check the following references for completeness: 16, 62, 72, 78, 80 (numbers changed because Reference 35 and 36 were added as suggested by the Reviewer)

Response: Thank you for thoroughly revising the Reference list. The respective references were checked and corrected. The Pichia Expression Kit User Guide from Invitrogen is usually cited as we did. It can easily be found online.

Supplementary Figure S3: axis labels: decimal point not comma

Response: Adapted.

Supplementary Figure S4: Figure legend with different font sizes. Molecular weight marker lanes should be visible (and also, non-manipulated western blots should be provided, even if for review purposes only).

Response: The font sizes of the Figure legend were unified. Since immunoblot detection with HRP-labelled antibodies does not co-stain the protein ladded, we cannot provide visible molecular weight marker lanes, but instead label the size as observed on the blot. Given the deadline for resubmission (which was yesterday), we cannot provide the original western blot images at the time, because they are saved on the computer linked to the G-box at the Institute of Molecular Boiotechnology at TUGraz. The people who would be able to allocate them (Astrid Radkohl and Lukas Bernauer) will be on vacation until January 8th, which means that we could submit these later, if necessary.
